# Colossal tunability in high frequency magnetoelectric voltage tunable inductors

Yongke Yan [1,2], Liwei D. Geng[3], Yaohua Tan[4], Jianhua Ma[4], Lujie Zhang[5], Mohan Sanghadasa[6], Khai Ngo[5], Avik W. Ghosh[4], Yu U. Wang[3] & Shashank Priya[1,2]

The electrical modulation of magnetization through the magnetoelectric effect provides a great opportunity for developing a new generation of tunable electrical components. Magnetoelectric voltage tunable inductors (VTIs) are designed to maximize the electric field control of permeability. In order to meet the need for power electronics, VTIs operating at high frequency with large tunability and low loss are required. Here we demonstrate magnetoelectric VTIs that exhibit remarkable high inductance tunability of over 750% up to 10 MHz, completely covering the frequency range of state-of-the-art power electronics. This breakthrough is achieved based on a concept of magnetocrystalline anisotropy (MCA) cancellation, predicted in a solid solution of nickel ferrite and cobalt ferrite through first-principles calculations. Phase field model simulations are employed to observe the domain-level strain-mediated coupling between magnetization and polarization. The model reveals small MCA facilitates the magnetic domain rotation, resulting in larger permeability sensitivity and inductance tunability.

[1] Center for Energy Harvesting Materials and Systems, Virginia Tech, Blacksburg, VA 24061, USA. [2] Department of Materials Science and Engineering, Pennsylvania State University, University Park, PA 16802, USA. [3] Department of Materials Science and Engineering, Michigan Technological University, Houghton, MI 49931, USA. [4] Department of Electrical and Computer Engineering, University of Virginia, Charlottesville, VA 22904, USA. [5] Center for Power Electronics Systems (CPES), Virginia Tech, Blacksburg, VA 24061, USA. [6] Weapons Development and Integration Directorate, Aviation and Missile Research, Development, and Engineering Center, US Army RDECOM, Redstone Arsenal, AL 35898, USA. These authors contributed equally: Yongke Yan, Liwei D. Geng. Correspondence and requests for materials should be addressed to Y.Y. (email: yanthu@vt.edu) or to S.P. (email: sup103@psu.edu)

Electric field modulation of magnetic properties in magnetoelectric materials provides an opportunity to develop tunable fundamental circuit elements for realizing energy efficient electronics[1,2]. Progress has been made in understanding the electric field control of magnetic anisotropy[3], magnetic domain structure[4], spin polarization[5,6], etc. Electric field control of permeability in magnetic materials has led to a new class of magnetoelectric component, namely, voltage tunable inductors (VTIs)[7,8]. Inductance with large tunability offers a new paradigm for circuit design, especially for adaptive power conversion and tunable multi-band RF communication systems.

Figure 1a schematically illustrates the structure of magnetoelectric VTIs based on the strain-mediated magnetoelectric effect. A magnetoelectric VTI consisting of a piezoelectric and magnetic composite, operates on the principle of modulation of magnetic properties (permeability) in the magnetostrictive layer via interfacial stress induced by voltage-driven piezoelectric layers (Fig. 1b). In contrast to conventional magnetic components or devices, where the tunability is achieved either by mechanically tuning through bias permanent magnets or adjusting DC current through a control winding to saturate the iron core[9,10], magnetoelectric VTIs exhibit a smaller footprint, larger tunability and a lower energy consumption[1]. Various efforts have been made toward developing magnetoelectric VTIs, but the application of this new component is limited due to the constraint on high tunability and high cutoff frequency. For example, the highest inductance tunability reported in Metglas VTIs is up to 1150% at 1 kHz, but tunability decreases rapidly with increase in frequency due to high loss at high frequency (Blue line in Fig. 1d)[11]. The tunability of MnZn ferrites is about 56.6% at the frequency of 100 kHz, but drops to 20% at 1 MHz (Green line in Fig. 1d)[8]. For power electronic applications, the switching frequency of metal–oxide–semiconductor field-effect transistors (MOSFETs) is in the range 100 kHz to several MHz[12]. In future, the need for power electronics to have greater compactness and higher performance motivates the pursuit of higher switching frequencies. Therefore, the development of high-frequency tunable VTIs with large tunability and low loss is important.

Recently, we have conducted fundamental investigations on the effects of magnetocrystalline anisotropy, shape anisotropy, stress-induced anisotropy, and magnetic field bias induced anisotropy on the tunability of the magnetoelectric VTIs, suggesting a strategy for the design of magnetoelectric VTIs. Small magnetocrystalline anisotropy $K_1$ and low shape anisotropy $K_s$ are essential towards achieving large inductance tunability, where stress-induced anisotropy $K_\sigma$ becomes dominant over other anisotropies. For toroidal inductors where the shape anisotropy $K_s$ is absent, the tunability $\gamma$ is strongly dependent on the $K_\sigma / K_1$ ratio as shown in Fig. 1c. This design strategy led to the discovery of 1150% tunability in Metglas VTIs at low frequencies[11]. However, in general, low magnetocrystalline anisotropy will give rise to high permeability and low cut-off frequency according to Snoek's law, for an example, as observed in Metglas[13]. For high-frequency power electronics, Ni-Zn ferrite is one of most commonly used magnetic inductor materials. Ni-Zn ferrite has a high electric resistivity (>$10^{10}$ times higher than Metglas and >$10^4$ times higher than Mn-Zn ferrite) and low loss with working frequency from several MHz to a few hundreds of MHz depending on the

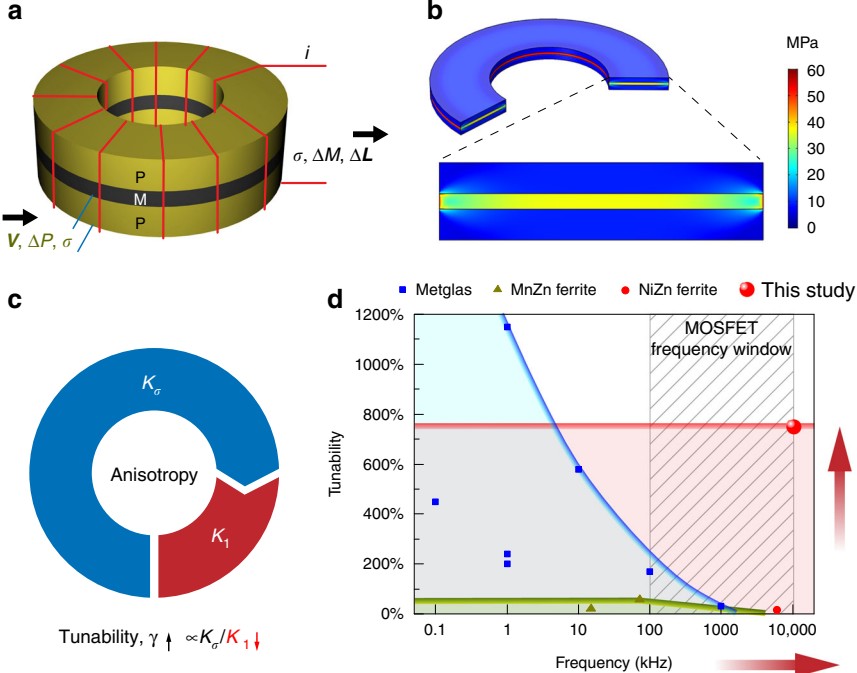

**Fig. 1** Magnetoelectric voltage tunable inductors (VTIs). **a** Schematic of the VTI structure. VTI operates on the principle of modulating magnetic properties (permeability) of the magnetostrictive layer via interfacial stress induced by voltage-driven piezoelectric layers. **b** Stress distribution in the VTI under 10 kV cm⁻¹. **c** Pie diagram of the magnetocrystalline anisotropy $K_1$ and stress-induced anisotropy $K_\sigma$ in a toroidal VTI core. The magnitude of change in the permeability in the magnetostrictive layer is strongly dependent upon the ratio of $K_\sigma/K_1$. Small $K_1$ is an essential requirement towards achieving large permeability and inductance tunability, $\gamma$. **d** Tunability vs. frequency of the state-of-the-art magnetoelectric VTIs. Blue square points: Metglas based VTI;[7,11] Green triangle points: MnZn ferrite based VTIs;[8] Red circle dot: NiZn ferrite VTIs;[11] Red sphere: this study. Frequency dependence of tunability of VTI with Metglas (Blue line[11]), MnZn ferrite (Green line[8]) and NiZn ferrite (Red line, this study). The lines show the frequency dependence of tunability for different types of VTI materials, while the data points are reported tunability at specific frequencies. In this study, minimized $K_1$ of the magnetoelectric VTIs via magnetocrystalline anisotropy cancellation between NiZnCu ferrite and $CoFe_2O_4$ provides a remarkable tunability of 750% up to 10 MHz, which fully covers the frequency range of state-of-the-art power electronics

composition[14,15]. However, the intrinsic magnetocrystalline anisotropy of NiZn ferrite is around $5000\,J\,m^{-3}$[16,17], which is 100 times larger than the value ($38\,J\,m^{-3}$) of Metglas[18]. The large magnetocrystalline anisotropy of NiZn ferrite based VTIs results in much lower tunability than Metglas VTIs[11]. Therefore, in order to design high-frequency magnetoelectric VTIs, a general approach is needed for significant reduction of the magnetocrystalline anisotropy of NiZn ferrite.

Here, we demonstrate a design principle for magnetoelectric VTIs that provides a direction for realizing high-frequency high-tunability VTIs. The design principle is based on the concept of magnetocrystalline anisotropy (MCA) cancellation and is demonstrated via synthesizing a solid solution of NiZnCu ferrite (NZCF) and $CoFe_2O_4$ ferrite. The compensation of the negative MCA of NZCF by the positive MCA of $CoFe_2O_4$ results in small or zero MCA, which gives rise to a remarkable high inductance tunability of 750% up to 10 MHz, completely covering the frequency range of state-of-the-art power electronics, as shown in Fig. 1d (Red line). First principles calculations using density function theory (DFT) was performed to guide the composition design based upon the MCA cancellation concept. Phase field model-based computer simulations were employed to observe the domain-level strain-mediated coupling between magnetization and polarization and elucidate the underlying mechanisms of electric field-dependent permeability. We believe that the scientific understanding provided in this study will provide a robust guideline for designing VTIs and allow the broader community to utilize the magnetoelectronic effect for inventing new circuit components.

## Results

**Magnetic properties of NCZF-CFO materials.** Figure 2a shows that, $\mu'$ the real part of permeability of NCZF-100xCFO, has strong dependence on the concentration of CFO. As the percentage of CFO increases, $\mu'$, continuously increases until reaching its maximum value at $x = 0.02$, and then starts to decrease monotonically. The imaginary part of permeability, $\mu''$, also exhibits a strong dependence on the CFO composition in the system NCZF-100xCFO, as shown in Fig. 2b. As the percentage of CFO increases, it can be observed that the magnitude of $\mu''$ peak continuously increases before reaching its maximum value at $x = 0.02$, and then starts to decrease with further increase in the CFO percentage. It is worth noting that the change of resonance frequency follows the Snoek's law, i.e., the material with high permeability $\mu'$ has low ferromagnetic resonance frequency (cut-off frequency). Figure 2c compares the permeability of NCZF-2CFO with the commercially available ferrite, Ferroxcube 4F1, which is widely used as the high-frequency inductor material. It can be observed that the cut-off frequency of NCZF-2CFO is similar to that of Ferroxcube 4F1. Figure 2d compares the loss factor of NCZF-100xCFO with Ferroxcube 4F1, where it can be noted that the addition of small amount of CFO will effectively reduce the loss of NCZF. The loss factor of NCZF-2CFO in the working frequency range is small and close to that of commercial Ferroxcube 4F1. It should be mentioned that the loss value of ferrite materials is sensitive to the composition, microstructure and processing condition variables (such as, sintering atmosphere, firing profile, etc.). The loss or energy consumption from piezoelectric layer[7] is negligibly small due to the high impedance of piezoelectrics. Building upon the results in Fig. 2 and prior study on loss factors, we focus on demonstrating a magnetoelectric VTI with large tunability through anisotropy engineering. Further, we provide complete understanding of the underlying strain-mediated magnetoelectric coupling mechanisms at the domain level.

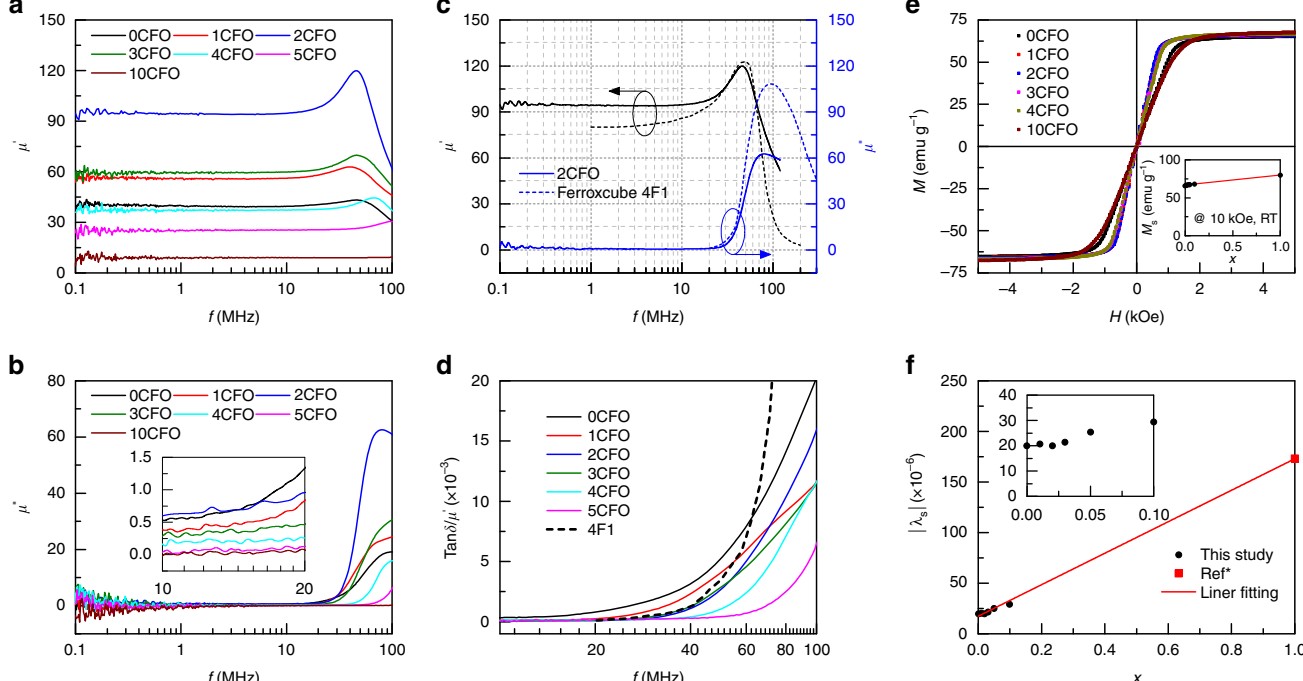

**Fig. 2** Magnetic properties of NCZF-CFO materials. **a** Frequency dependence of the real part of permeability, $\mu'$, of NCZF-100xCFO. **b** Frequency dependence of the imaginary part of permeability, $\mu''$, of NCZF-100xCFO. **c** Permeability comparison between NCZF-2CFO and commercial high-frequency ferrite Ferroxcube 4F1. **d** Loss factor $\tan\delta/\mu' = \mu''/(\mu'')^2$ comparison among different ferrite compositions. **e** Magnetization vs. magnetic field loops of NCZF-100xCFO. **f** Saturation magnetization $|\lambda_s|$ of NCZF-100xCFO. $|\lambda_s| = \frac{2}{3}(|\lambda_{11}| + |\lambda_{12}|)$, where $\lambda_{11}$ is the longitudinal magnetostriction measured when the strain gauge was parallel to the magnetic field, and $\lambda_{12}$ is the transverse magnetostriction measured when the strain gauge was perpendicular to the magnetic field. The data point of $|\lambda_s|$ for pure CFO ($x = 1$) refers to prior study (Ref. [24])

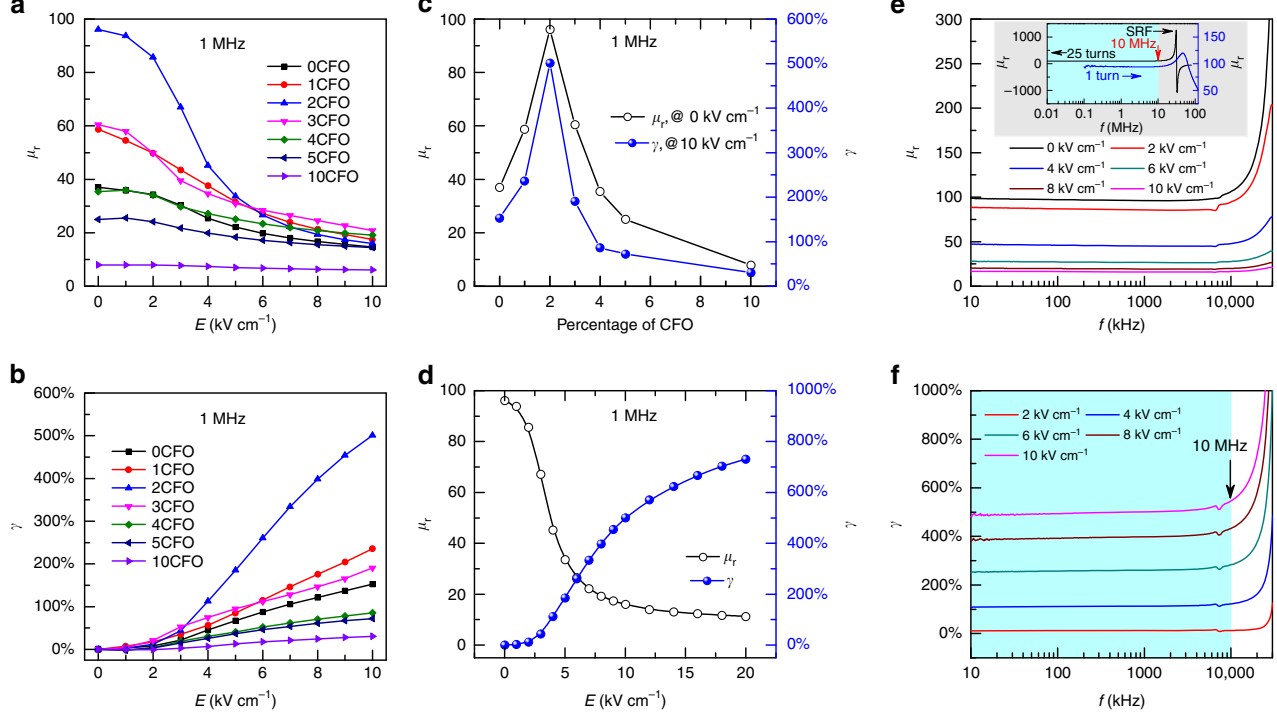

**Fig. 3** Tunability of magnetoelectric VTIs. **a** Permeability $\mu_r$ as a function of tuning electric field $E$ in NZCF-100$x$CFO VTIs, measured at 1 MHz. **b** Tunability $\gamma$ as a function of $E$ in NZCF-100$x$CFO VTIs, measured at 1 MHz. **c** $\mu_r$ and $\gamma$ as a function of the percentage of CFO in the NZCF-100$x$CFO solid solution. **d** $\mu_r$ and $\gamma$ as a function of $E$ in NZCF-2CFO VTI, measured at 1 MHz. **e** Frequency dependence of $\mu_r$ under different $E$, inset figure is the $\mu_r$ spectra for 25 turns and 1 turn winding coil under zero electric field. SRF: self-resonance frequency of inductance $L$ and parasitic capacitance $C$ circuit. **f** Frequency dependence of $\gamma$ under different $E$. $\mu_r$ and $\gamma$ show frequency stability up to 10 MHz

**Tuning behavior and frequency-dependent characteristics of magnetoelectric VTIs**. Figure 3a shows the permeability $\mu_r$ as a function of tuning electric field $E$ in NZCF-100$x$CFO VTIs measured at 1 MHz. It can be observed that the $\mu_r$ of all VTIs have strong field dependence and decrease with the increase in $E$. In order to quantify the tunability of $\mu_r$ with applied $E$, the tunability $\gamma$ is defined as $\gamma = (\mu_{r,0} - \mu_{r,E})/\mu_{r,E}$, where $\mu_{r,0}$ is the $\mu_r$ under zero electric field, and $\mu_{r,E}$ is the $\mu_r$ under a given electric field $E$. As shown in Fig. 3b, the $\gamma$ of all VTIs increases with increase in tuning electric field. The $\mu_r$ and $\gamma$ of VTIs have strong composition dependence as shown in Fig. 3c. The $\mu_r$ increases with the increase in the percentage of CFO and reaches maximum value at $x = 0.02$. With further increase in the percentage of CFO, the $\mu_r$ starts to decrease. The change of $\gamma$ shows a similar trend as $\mu_r$. Under tuning field of $E = 10$ kV cm$^{-1}$, the tunability $\gamma$ increases from 140% to 500% and then decreases rapidly. The maximum tunability $\gamma$ under $E = 10$ kV cm$^{-1}$ is ~500% for the composition of NZCF-2CFO. Figure 3d shows the electric field dependence of $\mu_r$ and $\gamma$ for a NZCF-2CFO VTI. The $\mu_r$ has a large tuning range from the initial value of 96 to 16 at $E = 20$ kV cm$^{-1}$, corresponding to the tunability $\gamma$ of 750%. Remarkably, the VTI with the composition of NZCF-2CFO not only has a remarkable large tunability, but also wide frequency stability. As shown in Fig. 3e, f, the $\mu_r$ and $\gamma$ of NZCF-2CFO VTI are almost constant up to the frequency of 10 MHz. With further increase in frequency, the $\mu_r$ and $\gamma$ further increase and approach self-resonance frequency (SRF) of 25 turns coiled inductor as shown in inset Fig. 3e.

**First principles calculation**. As shown above, the tunability of permeability and inductance in VTIs are strongly related to the content of CoFe$_2$O$_4$ in magnetic materials. It is well-known that while CoFe$_2$O$_4$ has large positive $K_1$ with easy axis along [001],

while NiFe$_2$O$_4$ has negative $K_1$ with easy axis along [111]. As depicted in Fig. 4a, magnetization reversal for the system with $K_1 = 0$ has no energy barrier because energy does not depend on the orientation of the magnetic moment, which gives a large magnetic susceptibility. First principles method is used to calculate the magnetic anisotropy $K_1$, magnetization $M_s$ and magnetostriction $\lambda_{100}$ of Ni$_{1-x}$Co$_x$Fe$_2$O$_4$. Figure 4b depicts the cation distribution of Fe and Co/Ni on the B-sites of the spinel structure used in the calculation. As shown in Fig. 4c, the calculated $K_1$ of NiFe$_2$O$_4$ and CoFe$_2$O$_4$ have different sign. The magnetocrystalline anisotropy is mainly due to the spin-orbit interaction, and the orbital motion of the electrons which couples with crystal electric field. The large difference of $K_1$ between NiFe$_2$O$_4$ and CoFe$_2$O$_4$ originates from the fact that by occupying the same octahedral site, the ground energy level of the orbital state for Ni$^{+2}$($5d^8$) is nondegenerate but Co$^{+2}$($5d^7$) is degenerate. Hence, the orbital magnetic moment for Ni$^{+2}$($5d^8$) is strongly quenched, while the orbital moment for Co$^{+2}$($5d^7$) is not fully quenched by the crystal field. The stronger spin-orbit coupling arises due to the unquenched orbital momentum of Co$^{+2}$($5d^7$) leading to a higher anisotropy in CoFe$_2$O$_4$[19,20]. Due to the opposite sign of $K_1$ in NiFe$_2$O$_4$ and CoFe$_2$O$_4$, the resultant anisotropy $K_1$ of Ni$_{1-x}$Co$_x$Fe$_2$O$_4$ solid solution will cross the zero value at $x \sim 0.05$ from linear prediction as shown in Fig. 4c. In this study, we introduced a small positive $K_1$ of CoFe$_2$O$_4$ into the negative $K_1$ of NZCF ferrite host in such a way that it minimizes the $K_1$ of ferrite, which is an essential for achieving large inductance and permeability tunability. It should be noted here that the difference of $x$ value for minimized $K_1$ between calculation ($x\sim0.05$) and experiment ($x\sim0.02$) may be due to the selection of initial calculation parameters and complexity of NZCF base composition (which is different from NiFe$_2$O$_4$). Figure 4d shows the magnetostriction $\lambda_{100}$ for Ni$_{1-x}$Co$_x$Fe$_2$O$_4$. The calculated values are $-41$ ppm for

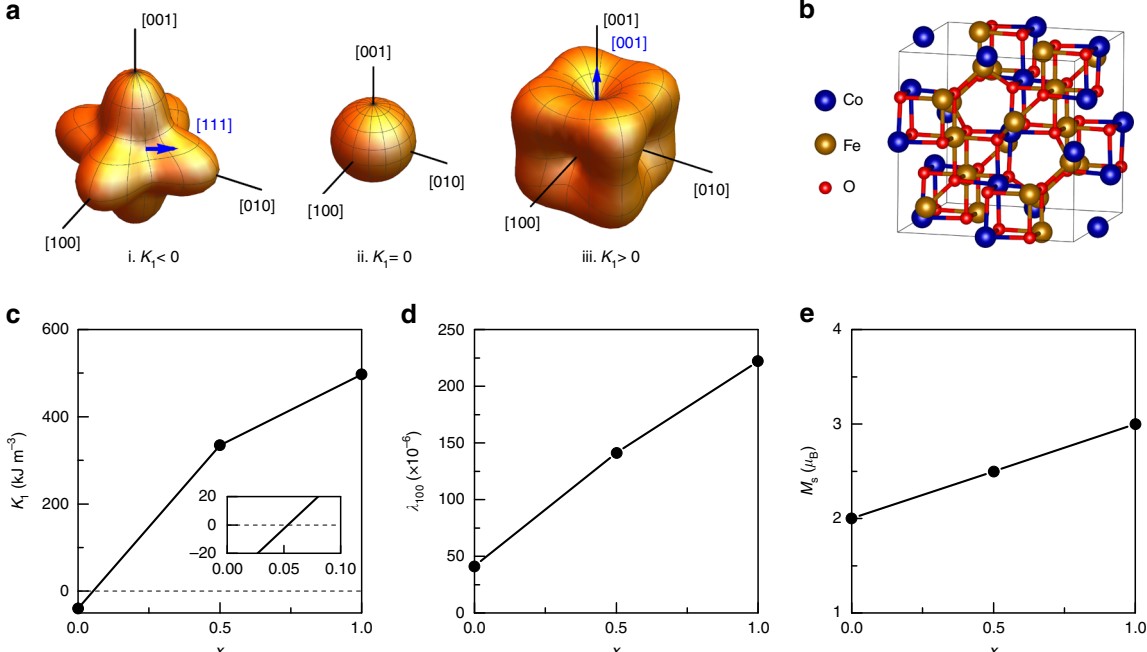

**Fig. 4** First principles calculation of NiFe$_2$O$_4$-CoFe$_2$O$_4$ solid solution. **a** Energy surfaces for cubic anisotropy with (i) $K_1 < 0$, (ii) $K_1 = 0$, (iii) $K_1 > 0$. Blue arrow shows the direction of easy axis. **b** Cation distribution of Fe (brown) and Co/Ni (blue) on the B-sites of the spinel structure (space groups *Imma*) used in the calculations. **c** Magnetocrystalline anisotropy $K_1$ of Ni$_{1-x}$Co$_x$Fe$_2$O$_4$; **d** Magnetostriction $\lambda_{100}$ of Ni$_{1-x}$Co$_x$Fe$_2$O$_4$; **e** Saturation magnetization per molecule (in $\mu_B$) of Ni$_{1-x}$Co$_x$Fe$_2$O$_4$

NiFe$_2$O$_4$ and −222 ppm for CoFe$_2$O$_4$, which agrees well with experimental data. The near linear behavior of $\lambda$ of Ni$_{1-x}$Co$_x$Fe$_2$O$_4$ indicates that the addition of 2% CoFe$_2$O$_4$ only slight increases the magnetostriction of NZCF, which agrees with the experimental results as shown in Fig. 2f. Figure 4e shows the saturation magnetization $M_s$ (in $\mu_B$) for Ni$_{1-x}$Co$_x$Fe$_2$O$_4$ system. Both Co$^{2+}$ and Ni$^{2+}$ are on the B site (octahedral sites) and form inverse spinel structure. The theoretical values of $M_s$ in CoFe$_2$O$_4$ and NiFe$_2$O$_4$ are 3 $\mu_B$ and 2 $\mu_B$, respectively. Similar to the $\lambda_s$, the linear behavior of $M_s$ of Ni$_{1-x}$Co$_x$Fe$_2$O$_4$ indicates that 2% CoFe$_2$O$_4$ addition into NZCF ferrite has negligible influence on the $M_s$ of magnetic materials as shown in Fig. 2e. In other words, based on the first principles calculation and experimental observation, the anisotropy cancellation effect via the combination of negative $K_1$ (NZCF) and positive $K_1$ (CoFe$_2$O$_4$) is the guiding principle for achieving the remarkably large permeability or inductance tunability in NZCF-2CFO.

**Phase field modeling of magnetic domain structures and tuning behavior.** To further elucidate the underlying mechanisms of VTI tuning behavior, phase field model-based computer simulations are employed. Results reveal the effects of voltage tunable piezoelectric strain on the permeability and its tunability for the laminate ferrite/PZT ME composite system. The phase field model developed in our previous work[21,22] was adopted, which explicitly addresses the domain-level strain-mediated coupling between magnetization and polarization. Since the MCA minimization or cancellation is crucial for the VTI to achieve high tunability, various values of MCA constant $K_1$ were varied in the range −10,000 to 10,000 J m$^{-3}$ to systematically investigate the MCA effect on VTIs. Figure 5a shows the magnetization evolution of the ferrite layer under electric field $E$ with different MCA constants $K_1$. At zero electric field, the simulations reveal that a larger $|K_1|$ leads to a smaller domain size due to the smaller exchange length. The magnetic permeability $\mu_r$ with various $K_1$ at $E = 0$ kV cm$^{-1}$ is plotted in Fig. 5b. It is noted that a larger MCA

results in a smaller permeability while a smaller MCA results in a larger permeability, which is in good agreement with experimental measurements shown in Fig. 3c. It should be noted here the microstructures of all NZCF-100$x$CFO ferrite have similar microstructure as shown in Supplementary Figure 1. The effect of microstructure variables (such as grain size, porosity) on the magnetic properties is not considered here. Since NZCF-2CFO possesses the largest permeability, it can be assumed that the $K_1$ for NZCF-2CFO is close to 0 J m$^{-3}$. In fact, regarding the non-uniformity which always exists in realistic materials, the MCA constant couldn't be homogenously zero in NZCF-2CFO solid solution, so the largest permeability observed in the experiments is much smaller than the simulated value.

When the tunable electric field is applied on the PZT layer along the thickness direction (Z-direction), an in-plane (XY-plane) compressive stress $\sigma$ will be exerted on the ferrite layer. Such a compressive stress will induce an effective magnetic uniaxial anisotropy, namely, the stress-induced anisotropy given by $K_\sigma = 3\lambda_s\sigma/2 < 0$ in the thickness direction. The negative stress-induced anisotropy would reduce the out-of-plane magnetization component and force higher volume of magnetizations to stay in the plane, as shown in Fig. 5a. The application of electric field $E = 200$ kV cm$^{-1}$ makes more magnetizations in favor of in-plane alignment. It is also noted that, a smaller MCA makes magnetization reorientation easier under the same electric field. Figure 5d shows the case of $K_1 = -2000$ J m$^{-3}$, where the permeability is significantly decreased by applying an electric field, associated with a large modulation of domain structures, resulting in a tunability of ~400% at $E = 400$ kV cm$^{-1}$. Such a behavior is consistent with experimental measurements shown in Fig. 3d. The tunability with various MCA constants $K_1$ at $E = 200$ kV cm$^{-1}$ is plotted in Fig. 5b, which reveals the same trend with the permeability, in good agreement with experimental measurements shown in Fig. 3c. Despite these agreements between simulations and experiments, the underlying mechanism on how the electric field reduces magnetic permeability and why a

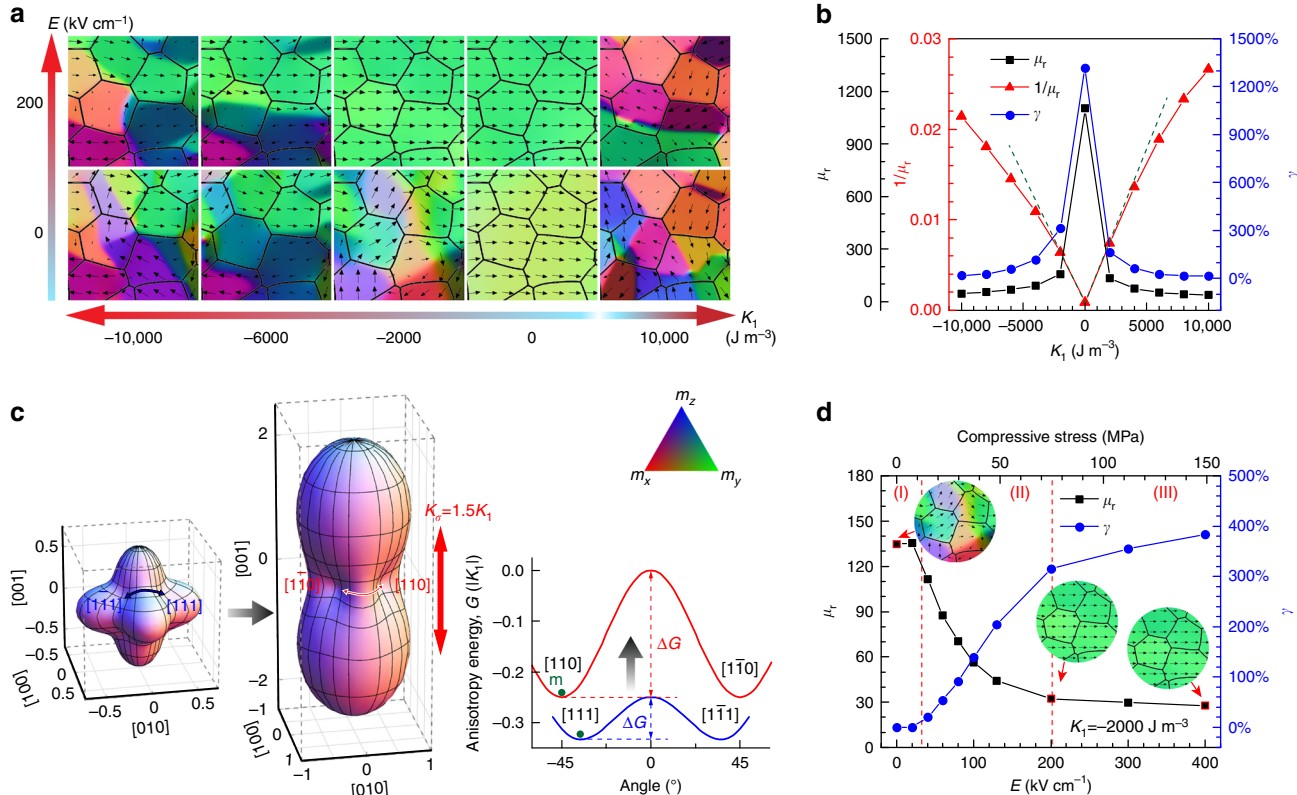

**Fig. 5** Phase field modeling of magnetization distribution and tuning behavior in magnetoelectric VTIs. **a** Magnetic domain structures of ferrite with different magnetocrystalline anisotropy $K_1$ under different electric field $E$. Black arrows represent magnetization distribution. Color contours indicate the three components of magnetization. **b** Magnetic permeability $\mu_r$ and its reciprocal $1/\mu_r$ at $E = 0$ kV cm$^{-1}$, as well as the tunability $\gamma$ at $E = 200$ kV cm$^{-1}$, with various MCA constants $K_1$. **c** Free energy contour for a single grain with cubic symmetry to illustrate the stress-induced anisotropy effect on the magnetization easy axis and domain rotation pathway, where both $K_1$ and $K_\sigma$ are negative. **d** Magnetic permeability $\mu_r$ and tunability $\gamma$ as a function of electric field $E$ for VTI with MCA constant $K_1 = -2000$ J m$^{-3}$. Inset figures are the corresponding magnetic domain structures of ferrite under the electric field $E = 0$ kV cm$^{-1}$, 200 kV cm$^{-1}$ and 400 kV cm$^{-1}$

smaller MCA results in a larger permeability or tunability still needs further explorations.

Magnetic permeability determines how easily the magnetization is reoriented under a magnetic field, and a small MCA usually enhances such magnetization reorientation and thus elevates the permeability. Specifically, two major processes contribute to the magnetization reorientation: domain rotation process and domain-wall motion process. Both processes manifest a strong dependency of MCA constant on magnetic permeability. Usually, $\mu_r \propto 1/|K_1|$ for the domain rotation process, while $\mu_r \propto 1/\sqrt{|K_1|}$ for the domain-wall motion process[18,22,23]. As such, the reciprocal of magnetic permeability $1/\mu_r$ with various $K_1$ at $E = 0$ kV cm$^{-1}$ is also plotted in Fig. 5b. It is noted that a linear relationship between $1/\mu_r$ and $K_1$ appears at small MCA, indicating that the domain rotation process is dominated when the MCA is relatively small, while an increased MCA begins to make $1/\mu_r$ deviate from linearity, indicating an increasing contribution from the domain-wall motion process. When an electric field is applied, the magnetization reorientation behavior will be modified. Since large tunability occurs at small MCA region in which the domain rotation process dominates, we mainly focus on the domain rotation process to investigate the tunability behavior under the influence of stress-induced anisotropy.

Figure 5c briefly illustrates the change in domain rotation pathway caused by the stress-induced anisotropy $K_\sigma$ in a single grain with cubic symmetry. With negative $K_1$, the magnetization is initially aligned along the easy axis direction < 111 >, and the

domain rotation process prefers to follow the pathway with the most minimized free energy. For example, under magnetic field applied in [010]-direction, the pathway $[111] - [101] - [1\bar{1}1]$ (Path I) is preferred since it possesses the lowest energy barrier $\Delta G$. When the electric field is applied, the negative stress-induced anisotropy would reduce the out-of-plane magnetization component, and the domain rotation pathway as well as the corresponding energy barrier would be also changed. As illustrated in Fig. 5c, the stress-induced anisotropy $K_\sigma = 1.5K_1$ is large enough to change the easy axis to < 110 > in-plane directions and make the domain rotation follow the new pathway $[110] - [100] - [1\bar{1}0]$ (Path II) whose energy barrier is much higher. Since domain rotation along Path II must overcome a higher energy barrier than along Path I, the magnetic permeability will be decreased. While we show results for a specific single grain in Fig. 5c, such a conclusion is also valid for the polycrystalline ferrite. Without stress-induced anisotropy ($E = 0$), domain rotation will follow the pathway with the largest reduction in energy. Once a large stress-induced anisotropy is induced ($E \gg 0$), the domain rotation will be constrained in the plane, which will make the in-plane pathway possess a higher energy and thus reduce the magnetic permeability. According to the above analysis, the permeability can be expressed in a general form

$$\mu_r = \chi + 1 = \frac{\mu_0 M_s^2}{2(K_0 + K_{0\sigma})} + 1, \qquad (1)$$

where $K_0$ represents the effective MCA, and the additional anisotropy $K_{0\sigma}$ represents the $\sigma$-induced anisotropy which depends on the stress-induced anisotropy $K_\sigma$. As illustrated in Fig. 5c, the $\sigma$-induced anisotropy $K_{0\sigma}$ will achieve its maximum when a high enough electric field is applied, which corresponds to a saturated minimum permeability.

According to Eq. (1), the stress-induced anisotropy will introduce an additional anisotropy which will decrease the permeability until it reaches the saturation state, which is consistent with our simulated results for the VTI with $K_1 = -2000 \, \mathrm{J \, m^{-3}}$ shown in Fig. 5d. By looking closely at Fig. 5d, three regimes can be defined. In Regime I, the permeability is initially increased at small electric field to reach a maximum, so that a small permeability peak ($\mu$–peak) emerges in this regime. Such a $\mu$–peak is also reported for inductor systems utilizing NiZn ferrite or other ferrite materials, and also appears in some of our experimental measurements as shown in Fig. 3a, usually attributed to the domain rotation process. In Regime II, the permeability drops fast because of the $\sigma$-induced anisotropy $K_{0\sigma}$, which is associated with a significant change of magnetic domain structures. In Regime III, the permeability decreases very slowly and approaches a saturated minimum value, because the anisotropy $K_{0\sigma}$ almost reaches its maximum due to the high electric field. It should be mentioned here that the range of electric field between simulations and experiments shows a big difference, which arises from the different values of $d_{31}$: the PZT structure adopted in the simulation has a much smaller $d_{31}$ than PMN-PT used in the experiment. According to Fig. 5d, as the electric field is tuned from 0 to $400 \, \mathrm{kV \, cm^{-1}}$, the induced compressive stress is changed from 0 to ~150 MPa, an appropriate stress range in comparison with the experiment.

Based on Eq. (1), assuming $\mu_r \gg 1$, the tunability can be given by

$$\gamma = \frac{\mu_{r0} - \mu_{rE}}{\mu_{rE}} \approx \frac{K_{0\sigma}}{K_0} \qquad (2)$$

Equation (2) indicates that, under the same electric field or stress-induced anisotropy, a smaller MCA corresponds to larger tunability, which explains why the large tunability occurs at small MCA constant $K_1$ under the same electric field, as shown in Figs. 3c and 5b. Furthermore, the anisotropy term $K_{0\sigma}$ not only includes the intrinsic anisotropy increment induced by $K_\sigma$ illustrated in Fig. 5c, but also the enhanced inhomogeneous stress from rough interfaces and grain boundaries under increased electric field or compressive stress, which also accounts for the tunability enhancement. It is worth mentioning that although the above analysis is based on domain rotation process, a similar analysis on the domain-wall motion mechanism also gives the same conclusion. We conclude that the tunability of ferrite/PZT ME composite VTIs can be enhanced by reducing the MCA constant $|K_1|$, which can be realized by means of alloying NiZn ferrite with a proper composition of CFO to effectively reduce $|K_1|$.

**Effect of internal bias stress on the tunability of VTI.** As observed in above experiments and simulations, three permeability regimes exist in the ME composite VTI system as shown in Fig. 5d. Since the tunability in Regime I is small or even negative, it is desirable to shift our initial working condition to avoid Regime I and operate the VTI directly in Regime II at $E > 0 \, \mathrm{kV \, cm^{-1}}$. Since magnetic permeability is directly controlled by the stress exerted on the ferrite layer, the working condition of VTI can be achieved by introducing a proper compressive internal bias stress on the ferrite layer, so that the permeability will be modulated in Regimes II and III under the same electric field range. Such a pre-existing internal bias stress can be introduced in different ways. Here, the internal bias stress is introduced by the electrical poling of the piezoelectric layer, so that the resulting compressive bias stress on the ferrite layer will shift the initial permeability into Regime II and then this VTI will work in Regimes II and III under electric field $E > 0 \, \mathrm{kV \, cm^{-1}}$. To achieve that goal, the piezoelectric layer should be unpoled before bonding with the ferrite layer.

Figure 6a, b show the experimentally measured permeability and corresponding tunability during the electrical poling process on the NZCF-2CFO/PMN-PT ME composite VTI system. In the first half cycle ($E = 0 \rightarrow 10 \, \mathrm{kV \, cm^{-1}}$), the permeability is almost constant at first (Regime I) and then suddenly drops at $E \sim 4 \, \mathrm{kV \, cm^{-1}}$ to a small magnitude without a big change by further increasing the electric field, indicating that the permeability falls into Regime III after the sudden drop. Such a permeability drop is due to the electrical poling of PMN-PT layer, which exerts a large compressive stress on the ferrite layer. In the second half cycle ($E = 10 \rightarrow 0 \, \mathrm{kV \, cm^{-1}}$), the permeability increases gradually and smoothly under decreased electric field, indicating that the permeability falls into Regimes II. The permeability $\mu_r \sim 40$ at $E = 0$ is much lower than that in the first half cycle $\mu_r \sim 90$, indicating the existence of a compressive internal bias stress. It is noted that because of the larger tuning range of strain / stress induced in the first half cycle, the tunability has a larger magnitude (~700%) than that (~300%) in the second half cycle. The permeability and tunability without internal bias stress (poled PMN-PT layer before bonding with ferrite) are also plotted in Fig. 6a, b. Comparison between the cases utilizing poled and unpoled (second half cycle) piezoelectric layer reveals that the introduction of internal bias compressive stress successfully shifts the magnetic permeability into Regimes II and III under the same tuning range of electric field.

Figure 6c shows the transverse strain hysteresis of piezoelectric ring in the aforementioned first and second half electric field cycles, which can help gain more insight into the permeability behavior. The sudden strain change $\Delta\varepsilon$ of $\sim -0.06\%$ occurs near the coercivity field $E_c \sim 4 \, \mathrm{kV \, cm^{-1}}$, which corresponds to a large compressive stress exerted on the ferrite layer to give rise to the significant decrease of permeability as shown in Fig. 6a. At $E = 0 \, \mathrm{kV \, cm^{-1}}$, the finite strain of approximately $-0.06\%$ remains, which corresponds to a finite compressive stress, and such stress is just the internal bias stress introduced by the electrical poling of PMN-PT layer. Once the polarizations are aligned along the poling direction, the poled state will not be destroyed unless a nonzero electric field in the opposite direction beyond the coercive field is applied.

To better understand the permeability behavior shown in Fig. 6a, associated domain structures obtained from the simulations for both ferrite and PZT layers at stage A (initial domain structure), stage B (nearly saturated domain structure), and stage C (irreversible domain structure at $E = 0 \, \mathrm{kV \, cm^{-1}}$) are shown in Fig. 6d. At initial stage A, both polarizations and magnetizations are almost randomly oriented due to zero internal stress at $E = 0 \, \mathrm{kV \, cm^{-1}}$. At stage B, ferroelectric polarizations are oriented uniaxially toward $Z$-direction under the high electric field, which induces an in-plane compressive stress on the ferrite layer, and the negative stress-induced uniaxial anisotropy forces the magnetizations to stay in the $XY$-plane. Since the polarizations are fully poled in $Z$-direction and magnetizations are well aligned in $XY$-plane, reducing electric field will not bring much difference to the domain structure between stage B and stage C. It is noted that at the same electric field $E = 0 \, \mathrm{kV \, cm^{-1}}$, Stage C exhibits a very different domain structure from Stage A, which is attributed to the compressive internal bias stress arising from the irreversible strain of piezoelectric layer.

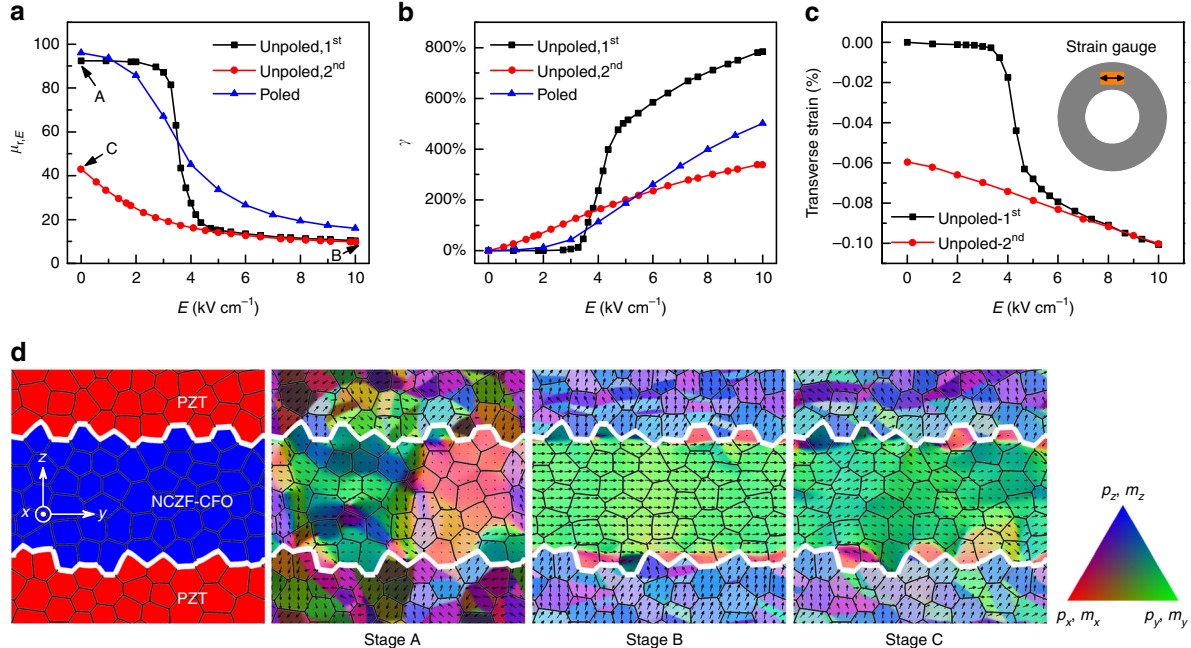

**Fig. 6** Effect of internal bias stress on the tunability of VTI. **a** Experimentally measured magnetic permeability $\mu_r$ under different electric fields $E$ for VTIs with initial unpoled piezoelectric layer and initial poled piezoelectric layer. Black line represents the unpoled sample with electric field increased from 0 to 10 kV cm$^{-1}$, red line represents the unpoled sample with electric field decreased from 10 to 0 kV cm$^{-1}$, while blue line represents the poled sample with electric field increased from 0 to 10 kV cm$^{-1}$. **b** Corresponding tunability $\gamma$ of permeability in **a**. **c** In-plane transverse strain of piezoelectric ring as a function of applied electric field, measured by strain gauge. **d** Simulated domain structures for both ferrite and PZT layers. Stage A, stage B and stage C correspond to the three stages as marked in **a**. Black arrows represent polarization or magnetization distribution. Color contours the three components of polarization or magnetization

In conclusion, we demonstrate a magnetoelectric VTI based on the concept of magnetocrystalline anisotropy (MCA) cancellation found in solid solution of NiZnCu ferrite and CoFe$_2$O$_4$ ferrite. The compensation of the negative MCA of NiZnCu ferrite by the positive MCA of CoFe$_2$O$_4$ results in small or zero MCA, which gives rise to a remarkable high inductance tunability of over 750% up to 10 MHz, completely covering the frequency range of state-of-the-art power electronics. Phase field model-based computer simulation were employed to depict the domain-level strain-mediated coupling between magnetization and polarization. The results indicate that the small MCA facilities the magnetic domain rotation, resulting in larger stress-permeability sensitivity. This study provides a guideline for designing high tunable inductor and allows the broader community to utilize the magnetoelectronic effect for inventing new circuit components.

## Methods

**Sample preparation**. The compositions of magnetostrictive ferrite ceramics are $(1-x)(Ni_{0.6}Zn_{0.2}Cu_{0.2})Fe_2O_4 - xCoFe_2O_4$ [abbreviated as NZCF-100xCFO], where $x = 0, 0.01, 0.02, 0.03, 0.04, 0.05$ and $0.10$. The ferrite powders were synthesized by the conventional solid-state reaction method using reagent-grade raw materials of NiO, ZnO, CuO, Fe$_2$O$_3$ and CoFe$_2$O$_4$. The mixture of raw materials was calcined at 800 °C for 4 h. Calcined powders were ball milled for 24 h, dried, and sieved. Then the sieved ferrite powder was pressed into annular shaped rings. These ferrite rings were finally sintered at 1050 °C for 2 h in air. The sintered ferrite rings were grinded and polished into the dimensions of outer diameter (OD) = 12.0 mm, inner diameter (ID) = 6.0 mm, and thickness ($t$) = 0.30 mm. The piezoelectric ceramic rings were prepared with similar procedure as ferrite ring. The composition of piezoelectric ceramics is Pb(Mg$_{1/3}$Nb$_{2/3}$)O$_3$–32.5PbTiO$_3$ [abbreviated as PMN-PT]. The PMN-PT powder was synthesized by the solid-state reaction method using reagent-grade raw materials of PbO, MgNb$_2$O$_6$, and TiO$_2$. The mixture of raw materials was calcined at 750 °C for 2 h. The PMN-PT powder was pressed into annular shape rings, followed by sintering at 1150 °C for 2 h in air. The sintered piezo rings were grinded and polished into the dimension of OD = 13.0 mm, ID = 6.0 mm, and thickness, $t$ = 0.60 mm. The polished piezo rings were electroded with silver paste (DuPont 7713) and fired at 550 °C for 30 min. After electroding, the piezo rings were poled along the thickness direction under electric

field of 30 kV cm$^{-1}$ for 5 mins at room temperature. To assemble a voltage tunable inductor core, a tri-layer composite structure consisting a ferrite ceramic ring sandwiched between two piezoelectric ceramic rings was bonded using epoxy (West System Epoxy 105 Resin and 206 hardener) and cured at room temperature for 24 h.

**Characterization**. For measurements of inductance and magnetic permeability, the toroidal inductors were fabricated by winding 25 turns of enameled copper wire around the piezo/ferrite laminated rings. The inductance and initial permeability were measured by using impedance analyzer (Keysight E4490A) at the frequency range of 10 kHz to 110 MHz. The tuning voltage on the piezoelectric rings was provided by a high DC voltage supply (Keithley 2290–5), which results in an electric field from 0 to 20 kV cm$^{-1}$ in the piezoelectric rings.

**First-principles calculations**. All calculations presented in this work were based on the projector-augmented wave PAW method which is implemented in the Vienna ab initio simulation package (VASP). A cutoff energy 800 eV and $k_{space}$ grid $7 \times 7 \times 7$ are used for all calculations for good convergence. For GGA + U calculation, we used $U_{eff} = U - J = 3$ eV for Fe, Co and Ni. We use the Pseudo-potential contributing 15 valence electrons per Co($3p^64s^23d^7$), 16 valence electrons per Ni ($3p^64s^23d^8$), 14 valence electrons per Fe($3p^64s^23d^6$), and 6 valence electrons per O ($2s^22p^4$). We first determined the equivalent lattice structure of CFO/NFO by relaxation based on GGA + U within collinear calculations. Then we determined the CFO/NFO geometries under symmetry breaking strains. Based on the relaxed structures, we performed non-collinear calculations to calculate magnetic anisotropy and magnetostriction. The detailed expressions and calculation are described in Supplementary Note 1.

**Phase field modeling and simulation**. In the phase field model, the ME composite system was described by field variables of magnetization $\mathbf{M}(\mathbf{r})$, polarization $\mathbf{P}(\mathbf{r})$, and free charge density $\rho(\mathbf{r})$. The evolutions of magnetization $\mathbf{M}(\mathbf{r}, t)$, polarization $\mathbf{P}(\mathbf{r}, t)$, and free charge density $\rho(\mathbf{r}, t)$ are respectively governed by the Landau-Lifshitz-Gilbert equation, the time-dependent Ginzburg-Landau equation, and microscopic Ohm's law. In order to systematically study the MCA effect on magnetic properties of the ME ferrite/PMN-PT composite VTI system, various values of magnetocrystalline anisotropy constant $K_1$ are considered in our simulations. Since the saturation magnetostriction constant $\lambda_s$ and magnetization $M_s$ vary only slightly with the small composition of CoFe$_2$O$_4$ in NiZnCu ferrite according to experimental measurements and first principles calculations, the values for both $\lambda_s$ and $M_s$ were fixed in our simulation cases. Since a complete set of

material parameters is not available for PMN-PZT, PZT is considered in the simulation study whose material parameters have been experimentally determined. More details on the phase field modeling and computer simulations can be found in Supplementary Note 2.

## Data availability

The data that support the findings of this study are available from the corresponding authors upon request.

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

## Acknowledgements

The authors gratefully acknowledge the financial support from DARPA MATRIX program, the NSF PFI-TT program (S.P.) and the NSF-Genome program NSF-DMREF-1235230.

## Author contributions

Y.Y. and S.P. conceived the idea. Y.Y. designed the experiment and conducted characterization. L.D.G. & Y.U.W. performed field phase modeling simulation. Y.T., J.M. and A.W.G. provided first principles calculation. L.Z. and K. N. advised electrical test and component design. M. S. and S.P. supervised the research. Y.Y., L.D.G., Y.U.W. and S.P. wrote the manuscript. All authors contributed discussion and revised the manuscript.

## Additional information

**Competing interests:** The authors declare no competing interests.

