## [Peer Review File · Nature Communications]

Reviewers' Comments:

Reviewer #1:

Remarks to the Author:

The authors used the composites of Co-ferrite and NiZn-ferrite for achieving a low magnetocrystalline anisotropy, which is needed for achieving a large voltage tunable inductance range for magnetoelectric inductors. It is well known that the Co-ferrites are lossy and are not a very good choice for power or radio frequency devices. At the same time, the authors did not show the magnetic loss tangents or the imaginary permeability spectrum, which are critical for such a magnetic material as a power inductor core. The authors also does not show any theoretical estimation of the expected voltage tunable inductance range, and compare with the obtained inductance range to see if the measured tunable inductance is reasonable. Also the authors did not show any quality factor of the inductors which are closely related to the imaginary permeability spectrum. In summary this paper lacks critical information needed for the claimed use as voltage tunable inductors.

Reviewer #2:

Remarks to the Author:

This manuscript presents a comprehensive work on the electric-field control of magnetic properties, based on the strain-mediated magnetoelectric coupling, which is helpful to the development of voltage tunable inductors. Both the experimental work and simulations are performed well and they agree with each other, validating the concept of optimizing the magnetoelectric coupling by engineering the magnetocrystalline anisotropy. I would love to know more about the device. How is the leakage of the device? How about the fatigue and degradation behavior after thousands to millions electric-field cycles? Do you worry about the Joule heating and breakdown issues when applying this device in real application? By the way, the thermodynamic potential of PMN-PT is already available. It will make the simulation more convincing by using the PMN-PT material parameters (PHYSICAL REVIEW APPLIED 8, 064005 (2017)) rather than the PZT parameters.

Reviewer #3:

Remarks to the Author:

The authors studied the strain-induced tunability in mixed ferrites of NZCF-xCFO. By tuning the composition of CFO, they have achieved the highest tunability over 750%. Theoretical modeling suggests that this pronounced feature strongly related to the magneto-anisotropy cancellation in the mixed compounds. However, the idea of this work is not novel. The same idea can be found in their previous work in Scientific Reports. Moreover, the value of tunability is not significantly higher than other reported data of VTI, although the experiment is convincing. In my opinion, this work do not meet the criteria of Nat. Comm. I think this work is more suitable for publishing on Scientific Reports after some amendments marked in the following. Ref: See i. g. Sci. Rep. 7, 16008 (2017), PRA 7, 044015 (2017), H. Lin, J. Lou, et al, IEEE Transactions on Magnetics 51, 1 (2015).

Marks:

p3. In Figure 1 d, it is not appropriate to draw a straight line (the red line) according to only one data point. Moreover, the other experiment data shown here is just lower than the highest one, which makes the trend line more confusing.

p6. As the definition of tunability γ is equal to $(\mu_{r,0} - \mu_{r,E}) / \mu_{r,E}$, the γ could represents the change of permeability under different electric

field, and further implies that the change mainly occurs at high fields. However, in Fig.2(a) and (b), the γ is rather following the trend of permeability at 0 kV/cm than $\mu_{r,E}$ at a higher electric field. That means at higher fields, the permeability among different samples are close to each other. Would the authors explain why these series ferrite materials have such behavior (the permeability are more different at low fields)?

p8. The authors claimed that the anisotropy and saturation magnetization are of extreme importance to determine the tunability of the mixed ferrites discussed here and they employed first principles calculation to estimate the two values. I think the results are lack of proper magnetometry or torque measurements to determine the anisotropy constant and saturation magnetization experimentally, in order to back their estimation. Could the authors try to do the related experiments?

p9. In the first principles calculation, the authors use NiFe₂O₄ instead of the true NZCF ferrites. In my opinion, the Zn and Cu are working as non-magnetic impurities, which could largely reduce the exchange stiffness of magnetic coupling, thus leading to a low anisotropy. The authors should show the difference between NiFe₂O₄ and NZCF to validate their simulation.

p10. Phase field modeling is employed in this manuscript to estimate the magnetization distribution in the ferrites. Here, I am curious about the domain size. As is well known, the domain size is proportional to $\sqrt{A/K}$, where A and K are the exchange constant and anisotropy energy. Since the K is almost cancelled (equals to zero) in the ferrites, the domain size could be very large. However, large domain size is usually not in favor of a small coercivity, which is another key parameter in the design of VTI and may harm the performance of VTI.

Reviewer #4:

Remarks to the Author:

This paper presents a very interesting and highly effective concept of electrical modulation of magnetization through the magnetoelectric effect that is promising for developing a new generation of tunable electrical components.

I am presently believe that the manuscript is probably interesting and relevant enough to be considered for Nature Communications, however at the same time, it is very fundamental problem I see about how different the approach described in this work form the original paper published by the same group in Sci Rep (Ref 9). In that previous work the same group of authors claimed they can do electric field modulation of magnetic properties via magnetoelectric coupling in composite materials used for magnetoelectric voltage tunable inductor and also showed extremely large inductance tunability of up to 1150% under moderate electric fields. Unless the authors would be able to strongly defend the novelty and originality of the approach presented here in this current manuscript there is still a major concerns: an expression "colossal tuning of magnetic permeability.." has to be thoroughly justified.

The authors also should comment on the potential efficiency (losses due to piezoelectric and magnetic elements) in VTI and the potential of performance restoring after several voltage tuning cycle (recovery , interface layer (glue) reliability , etc) , which is crucial for efficient repeatable functioning of the VTI devices.

Reviewers' comments:

Reviewer #1 (Remarks to the Author):

The authors used the composites of Co-ferrite and NiZn-ferrite for achieving a low magnetocrystalline anisotropy, which is needed for achieving a large voltage tunable inductance range for magnetoelectric inductors. It is well known that the Co-ferrites are lossy and are not a very good choice for power or radio frequency devices. At the same time, the authors did not show the magnetic loss tangents or the imaginary permeability spectrum, which are critical for such a magnetic material as a power inductor core. The authors also do not show any theoretical estimation of the expected voltage tunable inductance range, and compare with the obtained inductance range to see if the measured tunable inductance is reasonable. Also, the authors did not show any quality factor of the inductors which are closely related to the imaginary permeability spectrum. In summary this paper lacks critical information needed for the claimed use as voltage tunable inductors.

Answer: We agree with reviewer that loss is important factor for practical application. Pure Co-ferrite is a hard-ferrite and has higher hysteresis losses as inductor. In this study, only a small fraction of Co-ferrite was added to NiZnCu-ferrite to form solid solution. We have done this optimization carefully to minimize the losses while improving tunability and operational frequency range. Additional new experiments were conducted to quantify the losses and the results are shown in Figure 2. It can be found that the loss factor of our new VTI material (NCZF-2CFO) in the full working frequency range is small, quite similar to that of commercial Ferroxcube 4F1 (one of the most widely used high frequency inductor in power electronics). The theoretical estimation of tunability based on phase field modeling/simulations agrees well with the experimental results. The results validate the concept of optimizing the permeability tunability by engineering the magnetocrystalline anisotropy.

New Figure 2 | Magnetic properties of NCZF-CFO materials. **a**, frequency dependence of the real part of permeability, μ' , of NCZF-100xCFO. **b**, frequency dependence of the imaginary part of permeability, μ'' , of NCZF-100xCFO. **c**, Permeability comparison between NCZF-2CFO and commercial high frequency ferrite Ferroxcube 4F1. **d**, Loss factor $\tan\delta/\mu' = \mu''/(\mu')^2$ comparison among different ferrite compositions. **e**, Magnetization vs. magnetic field loops of NCZF-100xCFO. **f**, Saturation magnetization $|\lambda_s|$ of NCZF-100xCFO. $|\lambda_s| = \frac{2}{3}(|\lambda_{11}| + |\lambda_{12}|)$, where λ_{11} is the longitudinal magnetostriction measured when the strain gauge was parallel to the magnetic field, and λ_{12} is the transverse magnetostriction measured when the strain gauge was perpendicular to the magnetic field.

Reviewer #2 (Remarks to the Author):

This manuscript presents a comprehensive work on the electric-field control of magnetic properties, based on the strain-mediated magnetoelectric coupling, which is helpful to the development of voltage tunable inductors. Both the experimental work and simulations are performed well and they agree with each other, validating the concept of optimizing the magnetoelectric coupling by engineering the magnetocrystalline anisotropy. I would love to know more about the device. How is the leakage of the device? How about the fatigue and degradation behavior after thousands to millions electric-field cycles? Do you worry about the Joule heating and breakdown issues when applying this device in real application? By the way, the thermodynamic potential of PMN-PT is already available. It will make the simulation more convincing by using the PMN-PT material parameters (PHYSICAL REVIEW APPLIED 8, 064005 (2017)) rather than the PZT parameters.

Answer: Thanks for the encouraging comments. We provide some additional details:

- (1) *The leakage of the device:* Since the configuration of this magnetoelectric composite is 2-2 instead of 1-3, the leakage of VTI is related to the piezoelectric materials. PMN-PT has been widely used in design of piezoelectric devices, and the leakage is not found to be an issue. In this study, the leakage is found to be less than $1 \mu\text{A}$. Regarding the loss or energy consumption in tuning VTI, it is found to be negligibly small ($\sim 1 \text{ mJ}$) for power electronics (the power range of circuits such as converters and POL filters is typically from tens of watts to several hundred watts).
- (2) *The fatigue and degradation behavior:* The reviewer 4 also mentioned the reliability of device due to the existence of epoxy interface layer. In this study, our main goal was to demonstrate a new design rule for VTI materials to achieve high tunability (functionality) and high frequency range / broadband (practical application). The potential reliability issue of epoxy interface layer can be overcome via ceramic cofiring process, which totally avoids the use of epoxy. As shown in the figures below, we have already progress in this direction and developed a cofired multilayer VTI with good interface. These additional experiments were conducted for addressing the reviewer comments and will require more work to be published.

Cofired toroidal VTI – demonstration purposes only to address the reviewer comments.

Cofired solenoidal VTI – demonstration purposes only to address the reviewer comments.

- (3) *Joule heating and breakdown*: The Joule heating is not an issue since the piezoelectric layer has low leakage and works under static or off-resonance condition. Breakdown is also not an issue and technically it can be modified by using the approaches commonly deployed for commercial multilayer piezoelectric actuators and multilayer ceramic capacitors.
- (4) *Parameters for modeling*: We agree with reviewer that while it will be more convincing if using the recently reported PMN-PT material parameters, we think that the complete set of PZT parameters which have been determined experimentally provides good approximation for simulating the VTI behavior. We don't expect any major difference in the general trends of permeability/tunability presented in Figure 5 by using the PMN-PT material parameters.

Reviewer #3 (Remarks to the Author):

The authors studied the strain-induced tunability in mixed ferrites of NZCF-xCFO. By tuning the composition of CFO, they have achieved the highest tunability over 750%. Theoretical modeling suggests that this pronounced feature strongly related to the magneto-anisotropy cancellation in the mixed compounds. However, the idea of this work is not novel. The same idea can be found in their previous work in Scientific Reports. Moreover, the value of tunability is not significantly higher than other reported data of VTI, although the experiment is convincing. In my opinion, this work does not meet the criteria of Nat. Comm. I think this work is more suitable for publishing on Scientific Reports after some amendments marked in the following. Ref: See i. g. Sci. Rep. 7, 16008 (2017), PRA 7, 044015 (2017), H. Lin, J. Lou, et al, IEEE Transactions on Magnetics 51, 1 (2015).

Answers: This contribution and significance of this manuscript are different from our prior work published in Scientific Reports (Ref 9). As we mentioned in the introduction section that prior study was based on existing magnetic materials and goal was to investigate the effects of all material factors such as magnetocrystalline

anisotropy, shape anisotropy, stress induced anisotropy, and magnetic field bias induced anisotropy on the tunability of the magnetoelectric VTIs. Based on the study we presented guideline on how to select magnetic materials for design of VTI. That guideline led to the discovery of 1150% tunability in commercial Metglas VTIs **below 10 kHz frequency**.

However, even though the VTI based on commercial Metglas magnetic alloy has giant tunability of 1150%, it cannot be used for practical application because of very low cut-off frequency (less than 10 kHz) and high losses at high frequency (>1 MHz). It is not trivial to shift the operating frequency range from 10 kHz to 10 MHz (three orders of magnitude) without compromising the magnitude of tunability and increasing the losses. As mentioned in our prior work (Ref 9) and in introduction section of this manuscript, there is significant challenge in balancing the magnitude of tunability and operating frequency: Metglas based VTI has 1150% tunability but narrow frequency range (several kHz), while Ni-Zn ferrite, one of most commonly used magnetic inductor materials has wide frequency range (up to 6 MHz) but small tunability (16%). To overcome this fundamental challenge, discovery of new VTI material based on novel scientific principle is presented. We demonstrate for the first time VTI with high tunability (functionality) and high operating frequency range / broadband (practical application). We believe demonstration of tunability >750% up to 10 MHz will provide opportunity to implement VTI in power electronic circuits.

In order to avoid the confusion about novelty, we have modified the title as “Colossal tunability in high frequency magnetoelectric voltage tunable inductors”, which highlights the significance and contribution of this study. Please note colossal tunability is justified here as Ni-Zn ferrite based VTIs exhibit 16% tunability at 6 MHz while our new material exhibits 750% tunability at 6 MHz (about 46× increase).

Marks:

p3. In Figure 1 d, it is not appropriate to draw a straight line (the red line) according to only one data point. Moreover, the other experiment data shown here is just lower than the highest one, which makes the trend line more confusing.

Answers: I agree with the reviewer that the plot is slightly confusing. The line is drawn to show the frequency dependence of tunability for each type of material, and the data point is experimentally reported tunability for specific frequencies. We have added a note to clarify the representation in figure caption.

p6. As the definition of tunability γ is equal to $(\mu_r,0 - \mu_r,E) / \mu_r,E$, the γ could represents the change of permeability under different electric field, and further implies that the change mainly occurs at high fields. However, in Fig.2(a) and (b), the γ is rather following the trend of permeability at 0 kV/cm than μ_r,E at a higher electric field. That means at higher fields, the permeability among different samples are close to

each other. Would the authors explain why these series ferrite materials have such behavior (the permeability is more different at low fields)?

Answers: Such behaviors have been explained in the section “Phase field modeling of magnetic domain structures and tuning behavior” in this manuscript. The permeability is determined by magnetic anisotropy. There are two main types of material anisotropies contributing towards the performance of VTI, namely, magnetocrystalline anisotropy and stress-induced anisotropy. At low electric fields, magnetocrystalline anisotropy is dominant so the permeability magnitudes are different due to different magnetocrystalline anisotropy constants. At high electric fields, stress-induced anisotropy is dominant so the permeability values are quite close to each other since the electric-field-dependent stress-induced anisotropy is the same under the same electric field. Further increasing the electric field beyond a threshold will not have significant change on magnetic domain structures, which leads to almost saturated permeability (Regime III), as shown in Figure 5(d).

p8. The authors claimed that the anisotropy and saturation magnetization are of extreme importance to determine the tunability of the mixed ferrites discussed here and they employed first principles calculation to estimate the two values. I think the results are lack of proper magnetometry or torque measurements to determine the anisotropy constant and saturation magnetization experimentally, in order to back their estimation. Could the authors try to do the related experiments?

Answers: In this revised manuscript, we have included additional data (such as saturation magnetization and magnetostriction) as shown in new Figure 2. It can be concluded from this information that the enhanced tunability is not related to the magnetization and magnetostriction. The negative magnetocrystalline anisotropy of NiZn ferrite as well as the very large positive magnetocrystalline anisotropy of CoFe_2O_4 is well known based on the prior publications in literature. The new experimental results reported in the manuscript were validated through first principle calculations and field phase simulations. This not only validates the measurement but also provides comprehensive fundamental insights into the working principle of VTIs.

p9. In the first principles calculation, the authors use NiFe_2O_4 instead of the true NZCF ferrites. In my opinion, the Zn and Cu are working as non-magnetic impurities, which could largely reduce the exchange stiffness of magnetic coupling, thus leading to a low anisotropy. The authors should show the difference between NiFe_2O_4 and NZCF to validate their simulation.

Answers: We agree with the reviewer that including non-magnetic Zn and Cu impurities could lead to lower anisotropy. In doing so, very large supercells with too many configurations should be considered, which may result in an overloaded computation work but the impact will not be significant. Our main focus in this study is on revealing the relations between magnetic parameters (such as magnetocrystalline anisotropy, magnetostriction coefficient, and magnetic moment) and CFO composition x . To understand this point, NiFe_2O_4 is a good choice.

p10. Phase field modeling is employed in this manuscript to estimate the magnetization distribution in the ferrites. Here, I am curious about the domain size. As is well known, the domain size is proportional to $\sqrt{A/K}$, where A and K are the exchange constant and anisotropy energy. Since the K is almost cancelled (equals to zero) in the ferrites, the domain size could be very large. However, large domain size is usually not in favor of a small coercivity, which is another key parameter in the design of VTI and may harm the performance of VTI.

Answers: We agree with the reviewer that zero K will result in large domain size and may harm the performance of VTI. However, the intrinsic magnetocrystalline anisotropy K_1 will never be absolutely or uniformly zero in real ferrite materials due to the existence of grain boundaries and defects. Further, shape anisotropy and stress-induced anisotropy will also contribute towards the total anisotropy K . As shown in Figure 2, the VTI utilizing NZCF-2CFO ferrite possesses the smallest magnetocrystalline anisotropy and it still exhibits a low loss like other cases.

Reviewer #4 (Remarks to the Author):

This paper presents a very interesting and highly effective concept of electrical modulation of magnetization through the magnetoelectric effect that is promising for developing a new generation of tunable electrical components.

I am presently believing that the manuscript is probably interesting and relevant enough to be considered for Nature Communications, however at the same time, it is very fundamental problem I see about how different the approach described in this work form the original paper published by the same group in Sci Rep (Ref 9). In that previous work the same group of authors claimed they can do electric field modulation of magnetic properties via magnetoelectric coupling in composite materials used for magnetoelectric voltage tunable inductor and also showed extremely large inductance tunability of up to 1150% under moderate electric fields. Unless the authors would be able to strongly defend the novelty and originality of the approach presented here in this current manuscript there is still a major concern: an expression "colossal tuning of magnetic permeability." has to be thoroughly justified.

Answers: This contribution and significance of this manuscript are different from our previously published work in Scientific Reports (Ref 9). As we mentioned in the introduction section that prior study was based on existing magnetic materials and goal was to investigate the effects of all material factors such as magnetocrystalline anisotropy, shape anisotropy, stress induced anisotropy, and magnetic field bias induced anisotropy on the tunability of the magnetoelectric VTIs. Based on the study we presented guideline on how to select magnetic materials for design of VTI. That guideline led to the discovery of 1150% tunability in commercial Metglas VTIs **below 10 kHz frequency**. However, even though the VTI based on commercial Metglas magnetic alloy has giant tunability of

1150%, it cannot be used for practical application because of very low cut-off frequency (less than 10 kHz) and high losses at high frequency (>1 MHz). It is not trivial to shift the operating frequency range from 10 kHz to 10 MHz (three orders of magnitude) without compromising the magnitude of tunability and increasing the losses. As mentioned in our prior work (Ref 9) and in introduction section of this manuscript, there is significant challenge in balancing the magnitude of tunability and operating frequency: Metglas based VTI has 1150% tunability but narrow frequency range (several kHz), while Ni-Zn ferrite, one of most commonly used magnetic inductor materials has wide frequency range (up to 6 MHz) but small tunability (16%). To overcome this fundamental challenge, discovery of new VTI material based on novel scientific principle is presented. We demonstrate for the first time VTI with high tunability (functionality) and high operating frequency range / broadband (practical application). We believe demonstration of tunability >750% up to 10 MHz will provide opportunity to implement VTI in power electronic circuits. In order to avoid the confusion about novelty, we have modified the title as “Colossal tunability in high frequency magnetoelectric voltage tunable inductors”, which highlights the significance and contribution of this study. Please note colossal tunability is justified here as Ni-Zn ferrite based VTIs exhibit 16% tunability at 6 MHz while our new material exhibits 750% tunability at 6 MHz (about 46x increase).

The authors also should comment on the potential efficiency (losses due to piezoelectric and magnetic elements) in VTI and the potential of performance restoring after several voltage tuning cycle (recovery, interface layer (glue) reliability, etc.), which is crucial for efficient repeatable functioning of the VTI devices.

Answers:

(1) *Potential efficiency*: Regarding the loss or energy consumption from piezoelectric layers, as mentioned in a previous study [Applied Physics Letters 94, 112508 (2009)], it is extremely small [~ 1 mJ] due to the high impedance of piezoelectrics. This magnitude is negligible for power electronics where the power range is typically from tens of watts to several hundred watts. Regarding the loss from magnetic parts, we have added new experimental results (new Figure 2C and D, see the figure below). The loss factor of NCZF-2CFO in the working frequency range is small, and close to commercial Ferroxcube 4F1 (one of the most widely used high frequency inductor for power electronics).

New Figure 2 | Magnetic properties of NCZF-CFO materials. **a**, frequency dependence of the real part of permeability, μ' , of NCZF-100x CFO . **b**, frequency dependence of the imaginary part of permeability, μ'' , of NCZF-100x CFO . **c**, Permeability comparison between NCZF-2CFO and commercial high frequency ferrite Ferroxcube 4F1. **d**, Loss factor $\tan\delta/\mu' = \mu''/(\mu'')^2$ comparison among different ferrite compositions. **e**, Magnetization vs. magnetic field loops of NCZF-100x CFO . **f**, Saturation magnetization $|\lambda_s|$ of NCZF-100x CFO . $|\lambda_s| = \frac{2}{3}(|\lambda_{11}| + |\lambda_{12}|)$, where λ_{11} is the longitudinal magnetostriction measured when the strain gauge was parallel to the magnetic field, and λ_{12} is the transverse magnetostriction measured when the strain gauge was perpendicular to the magnetic field.

(2) *Reliability*: The interface layer (glue) reliability is indeed crucial for efficient and repeatable functioning of the VTI devices. However, this is not an issue when designing the practical manufacturing process. Ceramic cofiring process used for making parts in large quantities avoids the use of glue interface layer. As shown in the figures below, we have already successfully cofired multilayer VTI with good interfaces.

Cofired toroidal VTI – demonstration purposes only to address the reviewer comments.

Cofired solenoidal VTI – demonstration purposes only to address the reviewer comments.

Reviewers' Comments:

Reviewer #1:

Remarks to the Author:

The authors have addressed my comments. I therefore suggest that this manuscript be accepted.

Reviewer #2:

Remarks to the Author:

The authors have addressed all my questions and concerns. I suggest it be published as it is.

Reviewer #3:

Remarks to the Author:

The manuscript has been revised with some essential updates, and the title reflects the highlights better in the current form. However, the critical problem of this article remains on the innovation, which is to make it different to their previous work (Ref. 9). Although the authors has tried hard to emphasize the novelty in the frequency range, the basic concepts and designs are similar. Nevertheless, the current work is still a good instance of their previous method.

Reviewer #4:

Remarks to the Author:

It looks the authors addressed most of the comments, however, still they were not so convincing about the novelty in their approach as compared to their previous work , even though they showed a large inductance tunability (1150%) under moderate electric fields in a large frequency range. Their response showed that that they only provided a strategy for the designing of the device and to justify a choice of materials for the VTI and they agreed that the general principle and physics of the magnetoelectric coupling-based tuning have been already reported in Ref 9. Again I still have my reservations to accept the author's statement on novelty of the approach presented here and the title claiming "colossal tuning of magnetic permeability." as they called it a " guideline on how to select magnetic materials for design of VTIs", even though taking to account the technological importance of the paper and the results demonstrating VTIs performance.

I believe at the same time the authors addressed the losses and reliability issues adequately, and their approach seemed to be acceptable.

Reviewers' comments:

Reviewer #1 (Remarks to the Author):

The authors have addressed my comments. I therefore suggest that this manuscript be accepted.

Answers: We appreciate for the reviewer's consideration and encouraging comments.

Reviewer #2 (Remarks to the Author):

The authors have addressed all my questions and concerns. I suggest it be published as it is.

Answers: We appreciate for the reviewer's consideration and encouraging comments.

Reviewer #3 (Remarks to the Author):

The manuscript has been revised with some essential updates, and the title reflects the highlights better in the current form. However, the critical problem of this article remains on the innovation, which is to make it different to their previous work (Ref. 9). Although the authors have tried hard to emphasize the novelty in the frequency range, the basic concepts and designs are similar. Nevertheless, the current work is still a good instance of their previous method.

Answers: We appreciate the reviewer comments on our prior paper published in Scientific Reports (Ref 9). One fundamental question in design of tunable magnetostrictive materials is "how to approach the composition and microstructure design to provide near-zero magnetocrystalline anisotropy?" Without addressing this fundamental question, it is difficult to tailor the physical behavior of magnetoelectric composites with intended tunability. Based on our literature review, we believe this is the first study to propose and validate the concept of anisotropy cancellation for development of magnetostrictive materials with large stress-dependent tunability of permeability. Using this validated concept, we were successful in achieving high tunability at high frequency with low loss. We believe this strategy provides a general method that will be utilized by broader community in design of new class of materials. Theoretical model and simulations provided in this study will establish the knowledgebase needed for advancement of not only magnetostrictive materials but other classes of smart materials.

Reviewer #4 (Remarks to the Author):

It looks the authors addressed most of the comments, however, still they were not so convincing about the novelty in their approach as compared to their previous work, even though they showed a large inductance tunability (1150%) under moderate electric fields in a large frequency range. Their response

showed that that they only provided a strategy for the designing of the device and to justify a choice of materials for the VTI and they agreed that the general principle and physics of the magnetoelectric coupling-based tuning have been already reported in Ref 9. Again, I still have my reservations to accept the author's statement on novelty of the approach presented here and the title claiming "colossal tuning of magnetic permeability." as they called it a " guideline on how to select magnetic materials for design of VTIs", even though taking to account the technological importance of the paper and the results demonstrating VTIs performance. I believe at the same time the authors addressed the losses and reliability issues adequately, and their approach seemed to be acceptable.

Answers: Thanks for the comments and suggestions. We agree with the reviewer that the electronic component being addressed “voltage tunable inductor” (VTI) is similar to that of Ref. 9. However, this study provides fundamental advancement towards design of VTIs by providing material design criterion that can yield large tunability over wide range of frequency. Without this material design criterion, prior studies have been based on simple strategy of combining either high magnetostrictive material or high permeability material with high piezoelectric constant material. But these studies had limited success in meeting all the requirements – tunability, bandwidth, loss, and co-firing capability. Our study provides VTIs that meet all these requirements and thereby provide transformative advancement in implementation of VTIs. At the same time, our material design strategy will be of interest of broader community in design of other classes of materials. Please note that the materials such as metglas (Ref. 9) cannot be used with industrial processes based on ceramic co-firing. The compositions identified in our study are completely compatible with mass manufacturing technique such as co-firing.